# Effects of Calcium Carbonate Microcapsules and Nanohydroxyapatite on Properties of Thermosensitive Chitosan/Collagen Hydrogels

**DOI:** 10.3390/polym15020416

**Published:** 2023-01-12

**Authors:** Premjit Arpornmaeklong, Natthaporn Jaiman, Komsan Apinyauppatham, Asira Fuongfuchat, Supakorn Boonyuen

**Affiliations:** 1Department of Oral and Maxillofacial Surgery, Faculty of Dentistry, Thammasat University Rangsit Campus, Rangsit 12121, Thailand; 2Master of Science Program in Oral and Maxillofacial Surgery, Faculty of Dentistry, Thammasat University Rangsit Campus, Rangsit 12121, Thailand; 3National Metal and Materials Technology Center, National Science and Technology Development Agency, Khlong Luang 12120, Thailand; 4Department of Chemistry, Faculty of Science and Technology, Thammasat University Rangsit Campus, Rangsit 12121, Thailand

**Keywords:** bone substitutes, calcium carbonate microcapsules, controlled release, nanohydroxyapatite particles, quercetin, thermosensitive chitosan/collagen hydrogel

## Abstract

Thermosensitive chitosan/collagen hydrogels are osteoconductive and injectable materials. In this study, we aimed to improve these properties by adjusting the ratio of nanohydroxyapatite particles to calcium carbonate microcapsules in a β-glycerophosphate-crosslinked chitosan/collagen hydrogel. Two hydrogel systems with 2% and 5% nanohydroxyapatite particles were studied, each of which had varying microcapsule content (i.e., 0%, 1%, 2%, and 5%). Quercetin-incorporated calcium carbonate microcapsules were prepared. Calcium carbonate microcapsules and nanohydroxyapatite particles were then added to the hydrogel according to the composition of the studied system. The properties of the hydrogels, including cytotoxicity and biocompatibility, were investigated in mice. The calcium carbonate microcapsules were 2–6 µm in size, spherical, with rough and nanoporous surfaces, and thus exhibited a burst release of impregnated quercetin. The 5% nanohydroxyapatite system is a solid particulate gel that supports homogeneous distribution of microcapsules in the three-dimensional matrix of the hydrogels. Calcium carbonate microcapsules increased the mechanical and physical strength, viscoelasticity, and physical stability of the nanohydroxyapatite hydrogels while decreasing their porosity, swelling, and degradation rates. The calcium carbonate microcapsules–nanohydroxyapatite hydrogels were noncytotoxic and biocompatible. The properties of the hydrogel can be tailored by adjusting the ratio of calcium carbonate microcapsules to the nanohydroxyapatite particles. The 1% calcium carbonate microcapsules containing 5% nanohydroxyapatite particle–chitosan/collagen hydrogel exhibited mechanical and physical strength, permeability, and prolonged release profiles of quercetin, which were superior to those of the other studied systems and were optimal for promoting bone regeneration and delivering natural flavonoids.

## 1. Introduction

Teeth loss can lead to progressive alveolar bone resorption and insufficient bone support for dental implants and prostheses [1,2]. However, limited availability and morbidity associated with autologous bone grafts [3,4] have promoted extensive use of bone substitutes. To improve clinical outcomes, bone substitutes should function as bioactive matrices and delivery vehicles for biomolecules and stem cells [5]. Hydrogels are three-dimensional networks of cross-linked hydrophilic polymers that are insoluble but can absorb large amounts of fluid [6,7,8]. Thermosensitive hydrogels have gained broad interest as bone substitutes that can be easily applied in minimally invasive surgery [9,10]. Hydrogels can fill minor and irregular defects and function as bioactive matrices and delivery vehicles for growth factors and stem cell transplantation [11,12]. The β-glycerophosphate (bGP)-cross-linked thermosensitive chitosan/collagen hydrogel has shown great potential as an injectable hydrogel for stem cell encapsulation and bioactive molecule delivery for promoting bone regeneration [13,14,15,16]. Chitosan and collagen are well-known biomaterials owing to their biocompatibility and osteoconductivity [8,17]. The chitosan/collagen composite improves the mechanical and physical properties of the collagen matrix and exhibits a porous structure that supports growth and osteogenic differentiation of bone marrow stromal cells [17,18,19].

The properties of the thermosensitive chitosan/collagen hydrogel can be modified by adjusting the chitosan/collagen ratio and incorporating a bioceramic. Increasing the chitosan/collagen ratio results in increasingly well-defined porous structures [15], and incorporating β-tricalcium phosphate (bTCP) increases the mechanical and physical strength of the hydrogels [7,8,16]. Additionally, adding an inorganic phase, such as nanohydroxyapatite (nHA), into the hydrogel increases the number of nucleation sites for mineralization, increases the matrix for cell adhesion, and enables integration of the newly formed bone with the surrounding bone [20,21]. Furthermore, inorganic matrices can support the hydrogel’s controlled release functionality by increasing its protein absorption and facilitating gradual release of bone morphogenetic protein-2 (BMP-2) [16,22]. Furthermore, incorporating inorganic phases as microcapsules into hydrogels can enhance storage and release of bioactive components [23]. Calcium carbonate exhibits excellent biocompatibility and has been used in biomedical applications, such as bone regeneration materials and drugs [24]. Bioactive molecules can be easily adsorbed onto the surface of porous calcium carbonate, and the degree of absorption into the matrix determines the nature of its sustained release [25]. Additionally, calcium-based microparticles have been reported to act as physical crosslinks in hydrogel systems [26].

As oxidative stress and chronic inflammation can affect wound healing and bone regeneration [27], addition of an antioxidant or anti-inflammatory agent would reduce these reactions [28]. Plant flavonoids such as quercetin (QT) are antioxidant and anti-inflammatory biomolecules that promote mesenchymal stem cell growth and osteoblastic differentiation [29,30]. Thus, it would be advantageous to deliver flavonoids and natural compounds into skeletal defects to promote growth and differentiation of mesenchymal stem cells, including hydrogel-encapsulated cells [15,30,31].

This present study focused on a system of thermosensitive composite hydrogels serving as a bioactive matrix and delivery vehicle of QT to provide an osteoconductive matrix and porous structure to support bone regeneration. The bioactive properties of the chitosan/collagen hydrogel were enhanced by incorporating inorganic particles, nHA, CaCO_3_ microcapsules, and a plant flavonoid, QT, into the hydrogel. Further, nHA and CaCO_3_ enhanced the osteoconductive properties and reinforced the strength and stability of the hydrogel. The incorporated QT-impregnated CaCO_3_ microcapsules (QT-CaCO_3_) in the hydrogel matrix functioned as a reservoir of QT, in which the released flavonoid was encapsulated in the hydrogel and gradually released. The microcapsule–hydrogel complex would enable concurrent controlled release of multiple bioactive molecules Therefore, in this study, we aimed to prepare QT-CaCO_3_ microcapsules and improve the properties of thermosensitive nHA–chitosan/collagen hydrogels by adjusting the ratios of the nHA and CaCO_3_ microcapsules in β-glycerophosphate-crosslinked chitosan/collagen hydrogels to obtain functional bioactive and injected hydrogels to promote bone regeneration. Cell cytotoxicity studies and subcutaneous implantation were performed in a mouse model to evaluate the biocompatibility of the microcapsules and hydrogels.

## 2. Materials and Methods

### 2.1. Chemicals and Materials

Acetic acid, aluminum chloride, calcium chloride, carboxymethylcellulose (average molecular weight (MW) 90,000 with 0.7 carboxymethyl groups per anhydroglucose unit), chitosan (medium molecular weight, 75–85% deacetylated), dimethyl sulfoxide (DMSO, Hybri-Max™, sterile-filtered, BioReagent, suitable for hybridoma, ≥99.7%), ethyl alcohol (200 proof, ACS reagent, ≥99.5%), β-glycerophosphate (BioUltra, ≥99% (titration), cell culture grade), hydroxyapatite nanopowder (<200 nm particle size (BET), ≥97% synthetic), lysozyme (chicken lysozyme, 50,000 unit/mg), QT, sodium nitrite, and sodium hydroxide were obtained from Sigma-Aldrich (St Louis, MO, USA). Atelocollagen 1% (*w*/*v*) was purchased from Koken (Tokyo, Japan). A human fetal osteoblast cell line (hFOB 1.19) was obtained from the American Type Culture Collection (ATCC) (Manassas, VA, USA). Phenol red-free Dulbecco’s Modified Eagle Medium/Nutrient Mixture F-12 (DMEM/F12) culture medium and cell culture plates were purchased from Corning (Corning, NY, USA), and fetal bovine serum, Antibiotic Antimycotic solution, and phosphate-buffered saline (PBS) were provided by Gibco (Thermo Fisher Scientific, Waltham, MA, USA). The CellTiter 96 AQueous One Solution Cell Proliferation Assay (MTS) was purchased from Promega (Madison, WI, USA). Disposable syringes (3 mL and filter paper (no. 1) were obtained from NIPRO (Japan) and Watman (UK), respectively. Vicryl 4-0 sutures were purchased from Ethicon (Somerville, NJ, USA).

### 2.2. Preparation of Calcium Carbonate Microcapsules

Calcium carbonate microcapsules (CaCO_3_) were prepared as described in a previous study [32], in which a 0.33 M sodium carbonate solution was poured into an equal volume of 1 mg/mL carboxymethylcellulose under stirring. To incorporate QT (Sigma-Aldrich, St Louis, MO, USA) into the microcapsules, 100,000 ppm QT (100 mg/mL in DMSO) was added to the sodium carbonate/carboxymethyl cellulose solution at 1:500 of the total volume of the mixture. Subsequently, the obtained solution with a final QT concentration of 100 ppm and DMSO 1:1000 (*v*/*v*) was vigorously stirred and heated to a temperature of 40 °C, following which an equal volume of 0.33 M calcium chloride was quickly poured into the solution. A cloudy solution was obtained and left at room temperature (RT) for 30 min to allow precipitation. The resulting white slurry was washed in deionized (DI) water and then in ethyl alcohol (Sigma-Aldrich) by suspension and subsequently centrifuged at 3500 rpm at 4 °C for 10 min. The supernatant fluid was removed, and the slurry was stored at −80 °C and freeze-dried [32].

### 2.3. Preparation of Thermosensitive Calcium Carbonate Microcapsules–Nanohydroxyapatite–Chitosan/Collagen Hydrogel

The hydrogels were categorized into two systems: System 1 comprised the hydrogel with 2% (*w*/*v*) nanohydroxyapatite (nHA), and System 2 comprised the hydrogel with 5% (*w*/*v*) nHA. In each system, there were four study groups according to the concentration of calcium carbonate microcapsules (CaCO_3_) incorporated: 0%, 1%, 2%, and 5% (*w*/*v*). A thermosensitive 4:1 (*w*/*w*) β-glycerophosphate-crosslinked chitosan/collagen hydrogel was prepared as previously described [14,16]. Briefly, 10 mL of 1% (*w*/*v*) atelocollagen (Koken, Japan) was poured into 20 mL of 2% chitosan in 0.1 M acetic acid under stirring on ice for 5 min to obtain a 4:1 (*w*/*w*) chitosan/collagen hydrogel. Then, the mixtures of 56% (*w*/*v*) β-glycerophosphate (bGP), nHA, and CaCO_3_ microcapsules were dropwise added into the chitosan/collagen matrix to achieve 1%, 2%, or 5% (*w*/*v*) CaCO_3_–2%, 5%, or 10% (*w*/*v*) nHA–10% bGP-4:1 (*w*/*w*) chitosan/collagen hydrogels (CaCO_3_–nHA–chitosan/collagen hydrogels) according to the groups of study; the mixtures were then stirred for 5 min on ice until they became a homogenous viscous hydrogel (all chemicals were from Sigma-Aldrich).

Then, to measure pH levels and observe the sol–gel transition of the hydrogels, the 4 mL hydrogels were poured into 10 mL glass bottles to measure the pH levels on ice, after which the gelation times were observed at 37 °C using the test tube inverting method. Movement of the hydrogel was observed every 5 min. The gelation time was determined when the movement of the gel was not observed and the opacity of the hydrogels increased. To prepare the samples for further investigation, the hydrogels were loaded into 10 mL glass bottles, 3 mL disposable syringes (NIPRO, Ayutthaya, Thailand), and Teflon molds (1 × 1 cm and 2 × 2 cm), resulting in volumes of 4, 2, 0.8, and 3 mL, respectively. The hydrogels were incubated at 37 °C for 18 h [14,15,16].

### 2.4. Scanning Electron Microscopy and Energy Dispersive Spectroscopy

The microstructures of the freeze-dried microcapsules and hydrogels were observed using field-emission scanning electron microscopy (FE-SEM) (JSM 7800F, JEOL, Tokyo, Japan). The mineral content of the hydrogels was examined using energy dispersive spectroscopy (EDS) installed in the FE-SEM. The freeze-dried microcapsules and 8.5 × 3 mm hydrogel disks were sputter-coated with platinum-gold and examined under FE-SEM. The morphology and surface characteristics of the microcapsules and porous structures as well as the distributions of the nHA nanoparticles and CaCO_3_ microcapsules in the hydrogel matrix were examined [16,33].

### 2.5. Rheological Testing

Strain and frequency sweeps were performed to investigate the structural stability and strength of the three-dimensional structures of the hydrogels and to examine the effects of nHA and CaCO_3_ incorporation on the viscoelastic properties of the hydrogel. A strain sweep was conducted to investigate the response of the hydrogel to the shear strain and shear stress. A frequency sweep was performed to identify the structure of the hydrogel at the molecular and microscale levels based on its behavioral response to frequency changes. The tests were conducted in the linear viscoelastic region, where the magnitudes of the rheological properties were constant and unaffected by the shear strain or shear stress.

The samples were cylindrical hydrogel disks (diameter: 25 mm; thickness: 2 mm) incubated at 37 °C for 18 h. The investigations were performed using a controlled-stress rotational rheometer (Gemini 200 HR Nano, Malvern Instruments, Worcestershire, UK) with 25 mm parallel plates. For these tests, the temperature was maintained at 37 ± 2 °C using a Peltier plate. For the strain sweep test, the shear strain range was set at <0.01–10 strain unit or <1–1000% shear strain under a constant 1 Hz frequency to determine linear viscoelastic ranges and strain-dependent behavior of the samples. The linear viscoelastic range of each group, mostly at 0.01 strain unit or 1% strain, was selected for further investigation in a frequency sweep test of the newly loaded samples. A constant strain of 0.1 strain unit with a frequency of 0.01–100 Hz was applied to the samples. The storage modulus (G′), loss modulus (G″), and complex viscosity (η*) were recorded and analyzed using (Gemini 200 HR NANO software, Malvern Instruments) [34].

### 2.6. Mechanical Properties

The compressive strength of the hydrogels was tested using a texture analyzer (TexturePro CT V1.9 Build 35, Brookfield Engineering Labs, Middleborough, MA, USA) under unconfined conditions. Hydrogel disks (diameter: 10 mm, thickness: 10 mm) were placed under a Delrin plate and compressed with a small tare load of 0.01 N at a ramp speed of 20% strain per second until reaching 20% strain on the hydrogel. Subsequently, the stress, strain, and elastic modulus were recorded [14,16].

### 2.7. A Mercury Intrusion Porosimetry

Open porosity was examined using mercury intrusion porosimetry (MIP) (MicroActive AutoPore V9600 1.03, Micromeritics Instrument Corporation, Norcross, GA, USA). Three of the five × 3 mm freeze-dried hydrogel samples were subjected to the test [16].

### 2.8. Swelling Test

Weight gain percentages were used to characterize the water absorption capacity of the hydrogels. The hydrogel disks, 10 × 10 mm in size, were placed in 60 mm cell culture plates and washed with 5 mL distilled water. After completely removing the water, a filter paper was briefly placed on the side and top of the tilted hydrogel disks to absorb excess water. The initial weights of the hydrogels were recorded (W_0_). The hydrogels were then immersed in 10 mL of PBS and incubated at 37 °C for 1, 6, 24, and 48 h. After each incubation period, the hydrogel was washed with DI water, tapped with filter paper, and weighed (W_1_). The percentage weight gain was calculated using the following formula: % weight gain = (W_1_ − W_0_) × 100/W_0_ [16].

### 2.9. Degradation Test

The percentage of weight loss was calculated to determine the rate of hydrogel degradation. The hydrogel disks, 10 × 10 mm in size, were first placed in 60 mm cell culture plates and washed with 5 mL distilled water. After the water was completely removed, as described in the procedure for the swelling test, the hydrogels were weighed to obtain their initial weight (W_0_). The hydrogels were then incubated with 10 mL of 10,000 U/mL lysozyme at 37 °C for 15 d. Then, 24 h after incubation and every 3 days thereafter, the hydrogels were washed with 5 mL of DI water, weighed (W_1_), and then re-immersed in a fresh lysozyme solution. The percentage weight loss was calculated as follows: % weight loss = (W_0_ − W_1_) × 100/W_0_ [16].

### 2.10. Release Profile of Flavonoids

A total flavonoid content assay was performed to quantify the amount of QT released from CaCO_3_ microcapsules and CaCO_3_–nHA–chitosan/collagen hydrogels. For the microcapsules, 100 mg/sample of microcapsules was placed in 15 mL tubes and incubated in 2 mL of phenol red-free DMEM/F12 culture medium. For the CaCO_3_–nHA chitosan/collagen hydrogels, the hydrogels were added to 24-well cell culture plates (0.5 mL per well) and incubated at 37 °C for 18 h. Then, the hydrogels were washed with DI water and incubated with phenol red-free culture medium (DMEM/F12) in each well at 0.5 Â mL/well for 15 d. On day 1 and subsequently every third day for 15 days, the supernatants of the microcapsules and hydrogels were collected and replaced with fresh culture medium. The supernatants were stored at −20 °C for TFC analysis [15,33,35].

For TFC analysis, the aluminum trichloride method was applied to quantify the total flavonoid content using QT as a reference. Briefly, the supernatant samples were mixed with 5% (*w*/*v*) sodium nitrite in DI water and incubated in the dark for 5 min, followed by addition of 10% (*w*/*v*) aluminum chloride and 1M sodium hydroxide in DI water, mixed well, and kept in the dark for 5 min. The absorbance of the solutions of the sample (whose color ranged from yellow to dark brown) was then measured at 510 nm using a microplate reader (Varioskan Flash, Vantaa, Finland). Culture medium was used as a negative control. The concentrations of QT were specified using a standard curve of QT in the culture medium and expressed in units of micrograms (µg)/mL QT [36].

### 2.11. Cytotoxicity Test of the Hydrogel

To prepare the pre-incubated media, 2% and 5% nHA–chitosan/collagen hydrogels with 1%, 2%, and 5% CaCO_3_ microcapsules were added to a 6-well cell culture plate (2 mL/well) and incubated at 37 °C in 5% CO_2_ and 95% humidity for 18 h. The hydrogels were washed with DI water, and 2 mL of growth medium was added to each well and incubated for 48 h. Then, the incubated culture medium was collected as a pre-incubated culture medium for cell culture. The growth medium consisted of a phenol red-free DMEM/F12 culture medium (Corning), 10% fetal bovine serum, and 1% antibiotic–antimycotic (Gibco, Thermo Fisher Scientific, Waltham, MA).

For cell culture, a human fetal osteoblast cell line (hFOB 1.19) was cultured in culture media pre-incubated with hydrogels. Human FOB1.19 were seeded in a 96-well cell culture plate (Corning) at 3000 cells/well and cultured in a growth medium for 24 h. Thereafter, the medium was changed, and cells were cultured in hydrogel-pre-incubated culture media, according to the study groups, for 72 h. Then, a cell viability assay was performed using CellTiter 96™ Aqueous One Solution Cell Proliferation Assay (MTS) (Promega, Madison, WI, USA) following the manufacturer’s instructions, and absorbance was measured at 490 nm using a microplate reader (Varioskan Flash) [15,33]. Then, the percentages of cell viabilities of the samples in the culture media pre-incubated with the hydrogels (S) were calculated relative to a control group in a regular growth medium (Ctr) as follows: % cell viability = (OD_S_/OD_ctr_) × 100 [37].

### 2.12. Subcutaneous Implantation in Mice

Animal research was conducted in compliance with the Guide for the Care and Use of Laboratory Animals of the National Research Council of Thailand (NRCT) and guidelines from the Institute of Animal for Scientific Purpose Development (IAD). This study was approved by the University Animal Ethics Committee of the Institutional Animal Care and Use Committee of Thammasat University (IACUC) (permission number 014/2021). Seven 8-week-old male C57BL/6NJcl mice (body weight: 25–28 g.) (Nomura Siam International Co., Ltd., Bangkok, Thailand) were purchased. Under the specified pathogen-free housing system at the Laboratory Animal Center (accredited by the AAALAC International’s Council, Thammasat University, Thailand), mice were acclimatized for 7 days before surgery and received RO water ad libitum and were fed with a standard commercial rodent diet. Groups of samples included (A) CaCO_3_ microcapsules, (B) 2% CaCO_3_–2% nHA–chitosan/collagen hydrogel, and (C) sham operation without implantation. There were five samples per group in Groups A and B, and three samples in the sham group (Group C).

The mice were anesthetized with isoflurane, and carprofen (5 mg/kg) was subcutaneously (SC) administered as a preemptive analgesic. Subcutaneous pockets were created 2 cm apart on the left shoulder and right tight areas, with two pockets per mouse in six mice and one in one mouse. Then, the samples were randomly injected into the pocket (0.5 mL/pocket) using 1 mL disposable injection syringes (2 groups per mouse). Except for the seventh mouse, one pocket was created for each sample. Subsequently, cutaneous wounds were closed using vicryl 4-0 sutures (Ethicon, Somerville, NJ, USA). Postoperatively, the surgical wounds and implantation sites were observed daily for 7 days and later once a week. Mice received subcutaneous tramadol (20 mg/kg) injections every 12–24 h for 3–5 days depending on the level of distress based on Grimace scores [38]. The mice were euthanized four weeks after implantation via carbon dioxide asphyxiation.

Skin, consistency, and swelling of the implant sites were examined. Subsequently, the specimens, including the layers of subcutaneous tissue below and skin above the specimens, were harvested. The specimens were fixed in 4% paraformaldehyde, embedded in paraffin, and stained with hematoxylin and eosin (H&E) for histological examination [39,40].

### 2.13. Histopathological Analysis for Biocompatibility

Histological slides were digitally scanned (Panoramic 250, 3Dhistech, Budapest, Hungary) for histological examination. Foreign body tissue reactions, infiltration of inflammatory cells, connective tissue infiltration, residual implant materials, and the presence of small blood vessels were examined [41]. The evaluation was performed under three representative high-power fields (magnification, ×40) for each section and three sections per sample [39,40,41].

### 2.14. Statistical Analysis

The number of samples in each investigation ranged between three and five. Data are presented as mean ± SD. Statistical analyses were performed using SPSS for Windows version 22 on a personal computer. Data were described and tested for normality using the Kolmogorov–Smirnov test. When the distribution was normal, the data were compared between groups and different investigation times using one-way ANOVA and repeated measures ANOVA with post hoc, Scheffe’s, or Dunnett’s test, respectively. When the data distribution was not normal, the Kruskal–Wallis test for independent samples was applied for comparisons between groups, and the Friedman test for matched samples was employed for comparisons between times. Dunn–Bonferroni post hoc tests were used for multiple comparisons. Statistical significance was set at *p* < 0.05.

## 3. Results

### 3.1. Calcium Carbonate Microcapsules

The obtained calcium carbonate microcapsules were spherical particles with diameters ranging from 2 µm to 6 µm and had rough and porous surfaces. The QT-incorporated CaCO_3_ microcapsules exhibited a burst release of QT in the first hour, following a gradual release of QT, which significantly decreased by day 7 and remained low until day 14 (Figure 1).

### 3.2. Effects of the Nanohydroxyapatite on Microstructure and Porosity of the Thermosensitive Calcium Carbonate Microcapsules–Nanohydroxyapatite–Chitosan/Collagen Hydrogel

The groups used in this study were 2%, 5%, and 10% nHA–2% CaCO_3_–chitosan/collagen hydrogels. FE-SEM demonstrated that the nHA–CaCO_3_–chitosan/collagen hydrogels were porous, with an interconnected open pore structure (Figure 2A–C). An increase in the nHA content decreased the porosity of the hydrogels and increased the surface area of the pore walls. The EDS analysis demonstrated that the principal components of the hydrogel were carbon (C) and oxygen (O), followed by calcium (Ca) and phosphate (P), respectively. The percentage weight of Ca and P increased with increasing concentrations of nHA, mainly Ca, in the 2–5% nHA hydrogels (Figure 2D–F).

### 3.3. Mechanical Strength and Degree of Degradation of the Thermosensitive Calcium Carbonate Microcapsules–Nanohydroxyapatite–Chitosan/Collagen Hydrogel

The mechanical strength and degradation rates of the 2% and 5% nHA groups were similar (*p* > 0.05) and were significantly higher than those of the 10% nHA group (*p* < 0.05) (Figure 2G,H). Therefore, 2% and 5% nHA were selected for further studying the effects of nHA and 1–3% CaCO_3_ microcapsules on the hydrogel properties (Figure 3).

### 3.4. Effects of Calcium Carbonate Microcapsules on Sol–Gel Transition Time and Microstructure of the Thermosensitive Calcium Carbonate Microcapsules–Nanohydroxyapatite–Chitosan/Collagen Hydrogel

The study groups were 1%, 2%, and 5% CaCO_3_ in the 2% and 5% nHA hydrogels (six groups). The hydrogels were in the liquid stage at 4 °C and became solid and opaque white gels after incubation at 37 °C for 7–15 min, depending on the concentrations of the CaCO_3_ and nHA in the hydrogels. In the 2% nHA hydrogels with 1%, 2%, and 5% CaCO_3_, the setting times were in the 15–10 min range, and those for the 5% nHA were 10–7 min.

The calcium carbonate microcapsules (CaCO_3_) had a more substantial influence on the hydrogel microstructure than nHA. An increase in CaCO_3_ microcapsule content decreased the porosity and pore connectivity of the hydrogels (Figure 3A–D,I–L). The 5% nHA groups with 0%, 1%, and 2% CaCO_3_ exhibited a more defined porous structure than the 2% nHA groups (Figure 3I–L). The deposition of nanoparticles and microcapsules on the pore walls of the hydrogels was content-dependent. The pore walls appeared rough with irregular surfaces (Figure 3E–H,M–P). The increase in the amounts of Ca and P with integration of nHA and CaCO_3_ in the porous structure of the hydrogel was supported by the dispersion of nano- and micrometer-sized particles on the pore walls (Figure 3E–H,M–P).

### 3.5. Mercury Intrusion Porosimetry of the Thermosensitive Calcium Carbonate Microcapsules–Nanohydroxyapatite–Chitosan/Collagen Hydrogel

Mercury intrusion porosimetry (MIP) showed that the hydrogels were porous, with porosities of 2% and 5% nHA being comparable with the average porosity of 84.17 ± 2.49% and median 83.94% of the total volume (Figure 4A). The levels of permeability of the 1% CaCO_3_–2% nHA (28,000 ± 8000 mDarcy) and 1% CaCO_3_–5% nHA (29,000 ± 7000 mDarcy) groups were markedly higher than those of the 2% CaCO_3_ (9200 ± 400 mDarcy) and 5% CaCO_3_ groups (3620 ± 30 mDarcy) (Figure 4B).

### 3.6. Mechanical Strength

The compressive test revealed that the compressive strength of the 5% nHA hydrogel was significantly higher than that of the 2% nHA hydrogel group (*p* < 0.05). However, when CaCO_3_ was added to the 2% nHA and 5% nHA–CaCO_3_ hydrogels, the strengths of the two sets were not significantly different (*p* > 0.05). This was because the CaCO_3_ microcapsules significantly increased the compressive strengths of the 2% and 5% nHA hydrogels (*p* < 0.05). The compressive strength of the 5% CaCO_3_ was significantly higher than those of the 0%, 1%, and 2% CaCO_3_–2% and 5% nHA groups, respectively (*p* < 0.05). The average compressive strength of the 1% and 2% CaCO_3_–2% and 5% nHA groups was 4000 ± 200 Pa (Figure 4C).

### 3.7. Physical Properties of the Thermosensitive Calcium Carbonate Microcapsules–Nanohydroxyapatite–Chitosan/Collagen Hydrogel

The CaCO_3_–nHA–chitosan/collagen hydrogel exhibited morphological stability with minimal water absorption. Changes in the percentage of weight gain after immersion in PBS for 24 h were minimal. The presence of CaCO_3_ and nHA increased the structural strength of the hydrogels to retain the fluid. The nHA hydrogels without calcium carbonate exhibited weight loss during the 24 h incubation (−13.54 ± 6.15%). The average weight gain of the 5% nHA groups (5.09 ± 0.40%) tended to be higher than that of the 2% nHA groups (0.45 ± 1.02%) (*p* > 0.05) (Figure 5A).

Regarding the degree of degradation, the percentage weight loss of the CaCO_3_–nHA–chitosan/collagen hydrogels on day 28 ranged from 5% to 20% of the original weight. On day 28, the percentage of weight loss mirrored the CaCO_3_ percentage in the hydrogels, with the highest to lowest weight loss occurring as follows: 0% CaCO_3_–2% nHA > 2% CaCO_3_–2% nHA > 2% CaCO_3_–5% nHA, 1% CaCO_3_–2% nHA, 1% CaCO_3_–5% nHA > 5% CaCO_3_–2% nHA, and 5% CaCO_3_–5% nHA. Regarding the concentrations of nHA, the effects of nHA were subtle; incorporation of 5% nHA tended to decrease the degradation rate or increase the physical strength of the hydrogel. The degradation rate of 0% CaCO_3_–2% nHA was significantly higher than those of 5% nHA–0% CaCO_3_ and the other groups (*p* < 0.05), and 5% CaCO_3_ with 2% and 5% nHA exhibited the lowest levels of degradation (Figure 5B).

### 3.8. Rheological Characteristics of the Thermosensitive Calcium Carbonate Microcapsules–Nanohydroxyapatite–Chitosan/Collagen Hydrogel

The samples for rheological analysis were categorized into four systems: (1) chitosan/collagen hydrogel alone, (2) hydrogel with nHA alone (2% and 5% nHA), (3) 2% nHA with CaCO_3_ microcapsules (1%, 2%, and 5%), and (4) 5% nHA with CaCO_3_ microcapsules (1%, 2%, and 5%). Figure 6 shows the amplitude dependence average values of the storage (G′) and loss (G″) moduli of the representatives of each system, and Table 1 lists the rheological parameters, which are the storage (G′) and loss (G″) moduli in the linear viscoelastic region (LVR) (G′@ LVR and G″@ LVR, respectively) of the strain sweep test at a frequency of 1 Hz. The incorporation of 2% and 5% nHA in System 1 increased the magnitudes of the storage and loss moduli (G′ and G″) but did not affect the yield points (σ @ G′ = G″) of the nHA hydrogels (System 2). Systems 1 and 2 exhibited linear viscoelastic behavior throughout the range of the studied shear strain. The addition of CaCO_3_ to the nHA hydrogels (Systems 3 and 4) further increased the magnitudes of both G′ and G″ and also increased the yield points (σ @ G′ = G″) of the systems. Calcium-based microparticles have been reported to act as physical crosslinks in hydrogels [26]. Moreover, a stress-stiffening hump of G″ was observed when CaCO_3_ microcapsules were added to the nHA hydrogel (Systems 3 and 4). Such a stress-stiffening hump is characteristic of particulate gel [42]. CaCO_3_ microcapsules with a particle size of 2–6 µm may dominate the rheological behavior of these systems. The incorporation of more CaCO_3_ microcapsules slightly increased the G′ and G″ moduli of the CaCO_3_–nHA hydrogels but did not affect the yield points of the hydrogels in Systems 3 and 4. Furthermore, the composite hydrogel with 2% nHA and 1%, 2%, and 5% CaCO_3_ exhibited a high standard deviation of the rheological parameters, particularly the @ G = G values in the nonlinear viscoelastic region, which were not observed in the 5% nHA groups (Table 1). The higher moduli of the matrix (i.e., 5% nHA chitosan/collagen hydrogel) of such hydrogel composites would alleviate the locally inhomogeneous response of the particulates.

Figure 7 shows the frequency dependence average values of G′ and G″ of the representatives of each system; Table 2. shows the important rheological parameters according to the power law of the frequency sweep test at 1% strain. System 1 exhibited the property of a weak gel, in which the values of the power law exponent (n′ and n″) were greater than 0. Conversely, when nHA was present in the system (System 2), the hydrogel exhibited strong gel characteristics, in which the values of n′ and n″ increasingly approached 0, with an increasing percentage of nHA in the system. When CaCO_3_ microcapsules were added to the nHA hydrogels (Systems 3 and 4), the particulate gel characteristics of the hydrogels were enhanced, and the values of n′ and n″ tended to 0.

### 3.9. Control Release Property of the Thermosensitive Calcium Carbonate Microcapsules–Nanohydroxyapatite–Chitosan/Collagen Hydrogel

QT was steadily released from the CaCO_3_–nHA–chitosan/collagen hydrogel during the first 72 h, and peak release was observed at 24 h. The release gradually decreased to extremely low levels on day 7 and remained low on day 14. Except for the 1% CaCO_3_ group, the release was sustained at the same rate from day 3 to 14. On days 7 and 14, the levels in the 1% CaCO_3_ group were significantly higher than those in the 2 and 5% CaCO_3_ groups. Thus, the amount of QT released varies with the percentage of CaCO_3_ microcapsules in the hydrogel. During the first 72 h, the amount of QT released from the 5% CaCO_3_ group was the highest, followed by the 5% CaCO_3_–2% nHA, 2%, and 1% CaCO_3_ groups, in that order (*p* < 0.05) (Figure 8).

### 3.10. Levels of Acid–Base Balance (pH) and Cell Cytotoxicity of the Hydrogels

The pH tended to increase with increasing concentrations of CaCO_3_ and nHA. The levels of 0–5% CaCO_3_ with 5% nHA (7.6 ± 0.1) and 2% nHA (7.5 ± 0.1) were not significantly different (*p* > 0.05). Average pH levels of 0% CaCO_3_ were 7.4 ± 0.1, 1%, 7.5 ± 0.1, 2%, 7.6 ± 0.1, and that of 5% CaCO_3_ was 7.7 ± 0.1 (Figure 9A). Regarding cell cytotoxicity, the percentage of cell viability of every group was 89 ± 12% of the control in the growth medium, in which the levels of 2% (88 ± 8%) and 5% nHA (86.8 ± 16%) were not significantly different (*p* > 0.05). In the 5% nHA hydrogel system, the percentage cell viability of the 1% CaCO_3_–5% nHA hydrogel (106 ± 6%) was significantly higher than the 2% and 5% CaCO_3_–5% nHA groups (77 ± 6%) (*p* < 0.05) (Figure 9B).

### 3.11. In Vivo Biocompatibility Analysis of the Thermosensitive Calcium Carbonate Microcapsules–Nanohydroxyapatite–Chitosan/Collagen Hydrogel

Four weeks after implantation, the mice that survived the surgery showed no signs of distress or postoperative infection, and the wound healing was uneventful. Small, well-defined rubbery bumps were identified at the implant sites of the microcapsules and hydrogel groups under normal and movable skin (Figure 10I–III). In the sham group, H&E-stained sections showed normal histology of the skin as well as subcutaneous tissue comprising keratinized epithelium, loose connective tissue, and hair follicles, with no infiltration of inflammatory cells (Figure 10A–C). In the microcapsule group, some infiltration of chronic inflammatory cells and small blood vessels at the implant site was observed (Figure 10D–F). In the hydrogel group, examination at low magnification (10×) revealed a residual hydrogel surrounded by thin fibrous connective tissue layers without necrotic tissue (Figure 11A) and covered with normal skin with dense connective tissue and keratinized epithelium without the presence of inflammatory cells (Figure 11B). At higher magnification (40×), five to eight layers of loose fibrous connective tissue were found, and, at the connective tissue–implant interface, infiltration of small blood vessels with moderate levels of chronic inflammatory cells and few multinucleated giant cells were found (less than five cells per visual field) (Figure 11C,D). There was no evidence of severe chronic inflammation with dense avascular and acellular fibrous tissue (Figure 11).

## 4. Discussion

The chitosan/collagen composite matrix is noncytotoxic and similar to the components of the bone matrix [43], and the porous structure of the matrix supports growth and osteogenic differentiation of bone marrow stromal cells [15,17]. Here, we used bGP and temperature changes to induce sol–gel transition at 37 °C, as reported in previous studies [14,16]. The bonding strength of the functional groups and setting time could be adjusted by increasing the concentration of bGP [14,19,44]. However, the adjustment was limited by the increase in pH and toxicity of the hydrogel with the increase in bGP [15], nHA, and CaCO_3_. Therefore, to maintain a physiological pH of 7.4, 10% bGP was used in this study.

It has been reported that a 1–3% bTCP content in a thermosensitive chitosan/collagen composite hydrogel increases the mechanical and physical strength of the hydrogel [16]. In the current study, we further developed a thermosensitive chitosan/collagen composite hydrogel by incorporating nHA and CaCO_3_ microcapsules into a thermosensitive chitosan/collagen composite hydrogel to improve its mechanical strength and bioactivity and control its release properties. The noncytotoxicity and stoichiometric ratio of Ca/P and the nanocrystal size of nHA, similar to that of HA crystals in bones, would further promote osteoconductive and protein absorption properties of the hydrogel [22,45]. Based on EDS analysis, nHA was a source of calcium and phosphate that could promote mineralization of the bone matrix [45]. As incorporation of nHA and CaCO_3_ microcapsules improved the mechanical and physical strength and controlled the release property of the thermosensitive chitosan/collagen hydrogel for bone regeneration, our findings support those of previous studies in that the nHA increased the mechanical strength of the hydrogel and promoted integration of the newly formed bone with the host bone [20,21], and incorporation of CaCO_3_ microcapsules would further enhance the osteoconductivity of the hydrogels [24]. This is because CaCO_3_ is a source of calcium ions and functions as a nidus for initiating mineralization of the bone matrix of the newly formed bone or osteoid [24], and the microcapsules can store bioactive molecules and release them from the hydrogels in a controlled manner [25,44].

QT-CaCO_3_ microcapsules were prepared using a co-precipitation method in which QT was simultaneously mixed with an aqueous solution of CaCO_3_. This method allowed for homogenous distribution of QT in the microcapsules [46]. The burst release of QT from the microcapsules within the first hour of incubation indicated that QT could be entrapped within the porous structure of the microcapsules and physically bonded with the CaCO_3_ particles [47,48]. The results agree with a previous report that hydrogel encapsulation helped increase QT retention for a gradual release of bioactive molecules at the target sites [28,49]. The controlled release of QT doses in a microcapsule-dependent manner suggested that the microcapsules helped to store the hydrophobic molecules of QT in the chitosan/collagen matrix, where the polycationic groups of chitosan, such as amine (NH^+^), facilitated dissolution and release of hydrophobic flavonoids into the encapsulated cells and external environment [50,51]. Thus, incorporation of CaCO_3_ microcapsules enables controlled delivery of multiple bioactive molecules from the hydrogel.

Based on the improvement in the microstructure, porosity, and mechanical and physical strength of the hydrogels, 2% and 5% nHA hydrogels were selected to study the effects of the percentage of CaCO_3_ microcapsules on the properties of the hydrogels. Incorporation of nHA and CaCO_3_ significantly increased the compressive strength of the hydrogel and decreased its degradation rate. Swelling behavior suggested that the 5% nHA with 1–5% CaCO_3_ exhibited more robust three-dimensional structures that could maintain and absorb fluid to a greater extent than the 2% nHA groups. The advantage of slow degradation of the hydrogels at 5–10% in physiologic lysozyme is that the physical stability of the hydrogels supports formation and deposition of the extracellular matrix or osteoid on the hydrogel during the early stage of bone formation during the first 2–6 weeks [52]. Additionally, slow degradation supports slow and prolonged release of active molecules [53]. The sustained release from the 1% CaCO_3_ groups could be attributed to the strength of the three-dimensional structure of the hydrogel and the permeability of the hydrogel, which were higher than those of the other groups.

Increasing particle deposition on the pore walls further increased the mechanical strength of the hydrogel and facilitated cell adhesion on the hydrogels [54]. However, increasing CaCO_3_ deposition decreased the permeability of the hydrogels. Thus, it might hinder mesenchymal stem cell ingrowth and blood vessel infiltration into the inner structure of the three-dimensional matrix of hydrogels [54,55,56]. Evidently, the permeability of the 2% and 5% nHA with 1% CaCO_3_ was markedly higher than that of the groups with 2% and 5% CaCO_3_, which supports the observation from the SEM analysis that an increase in CaCO_3_ content decreased the inter-pore connectivity, which can adversely affect the permeability of the hydrogel [57]. Permeability of the scaffolds is an important property that supports inflow and outflow of fluid, nutrients, oxygen, and metabolic waste in the 3D structure, including infiltration of MSCs into the inner structure of the scaffolds [58]. Thus, as permeability is an essential factor determining the ability of the hydrogel to maintain moisture and support nutrients for cells in the 3D structure [59], increasing the CaCO_3_ concentrations would adversely affect the growth and regeneration capacity of MSCs within the scaffolds and hydrogel.

The geometry of the porosity, size, interconnectivity, and roughness of the pore wall influence permeability [60]; bone substitutes should have higher permeability than the surrounding bones to promote fluid transportation into the scaffolds and improve survival and growth of the cells in the inner structure of the scaffolds [61]. While the groups with 1% CaCO_3_ had the highest levels of permeability at 25,000–30,000 mDarcy (2.5–3.0 × 10^−11^ m^2^), the levels in the 2% CaCO_3_ groups at 5000–10,000 mDarcy (5–10 × 10^−12^ m^2^) were much lower than those in the 1% CaCO_3_ group. However, the permeability of the 1% and 2% CaCO_3_ groups was higher than that of the cancellous bone of the femur (4.5 × 10^−12^ m^2^) [61]. Therefore, 2% and 5% nHA with 1–2% CaCO_3_ exhibited appropriate permeability of the bone substitute and scaffolds for stem cell transplants that would promote bone formation in the inner area of the three-dimensional scaffolds. Based on its rheological properties, 5% nHA was determined to be a firm gel. Therefore 1% CaCO_3_–5% nHA hydrogel was a strong particulate gel and an appropriate composition for thermosensitive chitosan/collagen hydrogel for promoting bone regeneration.

Furthermore, the rheological properties of the hydrogel controlled the fluid transportation phenomenon and compressive pressure within the hydrogel [59]. The stress and strain of the hydrogel influence the internal pressure of the three-dimensional structure, which regulates the fluid flow in and out of the hydrogel [61]. The transport phenomenon and internal pressure of the 1% CaCO_3_–5% nHA–chitosan/collagen hydrogel would stimulate cell growth and bone regeneration of the encapsulated cells [54,56,62]. Strain sweep and frequency sweep tests demonstrated that the CaCO_3_ microcapsules increased the mechanical strength of the composite hydrogels. At the same time, nHA enhanced the strength of the three-dimensional structure of the hydrogel to support dispersion of CaCO_3_ within the matrices. Thus, a higher amount of CaCO_3_ microcapsules increased the shear stress at the yield point, as well as the morphological or dimensional stability of the hydrogel. It can be hypothesized that the hydrogel structure can be adjusted to respond to shear stress and strain. In contrast, the permeability of the hydrogel could be controlled by adjusting the amounts of nHA and CaCO_3_. The distributions of nHA and CaCO_3_ on the pore walls of the porous hydrogel support this hypothesis.

Another essential factor for applicability of the hydrogel as a bone substitute is its mechanical strength to withstand external compressive force while maintaining space and form for bone regeneration [54], such as in lateral augmentation of alveolar bones, where the bone substitutes must withstand compressive force from the tension force of the flap and local tissue [63]. The current hydrogel is still not sufficiently strong to withstand these forces alone. The average mechanical strength of the 2% and 5% nHA with 1–5% CaCO_3_ was 4451 ± 684 Pa, significantly lower than those of the trabecular and cortical bones at 0.1–16 MPa and 130–200 MPa, respectively [64]. Therefore, the current hydrogel is applicable for well-defined bony defects, such as augmentation of three-wall periodontal defects, with use of a guided bone regeneration membrane to protect the dimension of the augmentation [9,40].

Moreover, the thermosensitive nHA–CaCO_3_–chitosan/collagen hydrogel was noncytotoxic and biocompatible. CaCO_3_ microcapsules and hydrogels are bioinert, do not cause excessive tissue reactions, and stimulate only mild foreign body reactions (FBR) [65,66]. Thin fibrous layers, low numbers of chronic inflammatory cells, and multinucleated giant cells indicate a low level of FBR [41,66,67]. These findings show that hydrogels are suitable for clinical applications in bone regeneration.

## 5. Conclusions

In this study, QT-CaCO_3_ microcapsules were prepared and incorporated into an nHA–β-glycerophosphate–chitosan/collagen hydrogel. The porous hydrogels were deposited with micro- and nanoparticles of CaCO_3_ and nHA on the pore wall, and 2% and 5% nHA increased the mechanical and physical strength of the hydrogels. The CaCO_3_ microcapsules further increased the mechanical strength, physical stability, and viscoelasticity of CaCO_3_–nHA hydrogels. The hydrogels controlled release of QT, and the release levels were related to the QT-CaCO_3_ concentration in the hydrogels. The QT-CaCO_3_ hydrogels exhibited the potential to simultaneously control the release of multiple bioactive molecules. The hydrogels were biocompatible because they were noncytotoxic and caused mild FBR. The 5% nHA hydrogel system exhibited the characteristics of a strong particulate gel with a well-defined porous structure and, along with the 1% CaCO_3_ microcapsule, exhibited high permeability and prolonged release of QT. Therefore, the findings indicate that thermosensitive 1% CaCO_3_–5% nHA–β-glycerophosphate–chitosan/collagen hydrogel can be used for delivering flavonoids for promoting bone regeneration. Further studies should be performed to investigate the anti-inflammatory properties and bone regeneration capacity of thermosensitive QT-CaCO_3_–nHA–chitosan/collagen hydrogels, including application of hydrogels for mesenchymal stem cell encapsulation and transplantation in skeletal defects.

## Figures and Tables

**Figure 1 polymers-15-00416-f001:**
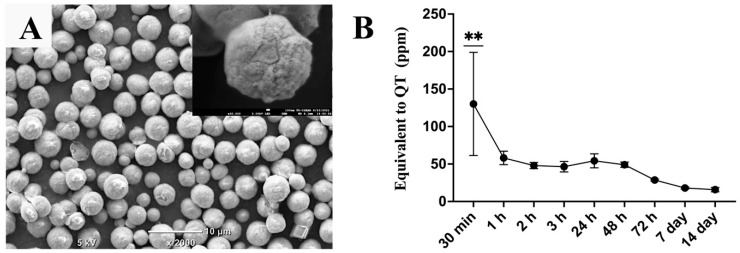
(**A**) Scanning electron microscopy image of the QT-incorporated calcium carbonate microcapsules. (**B**) Total flavonoid-content assay demonstrating a release profile of QT from the microcapsules incubated in phosphate buffer saline at 37 °C for 14 days; magnification bars represent 10 μm. The symbol ** denotes that the value is significantly higher than at other investigation times, with *p* < 0.01 (n = 5, mean ± SD).

**Figure 2 polymers-15-00416-f002:**
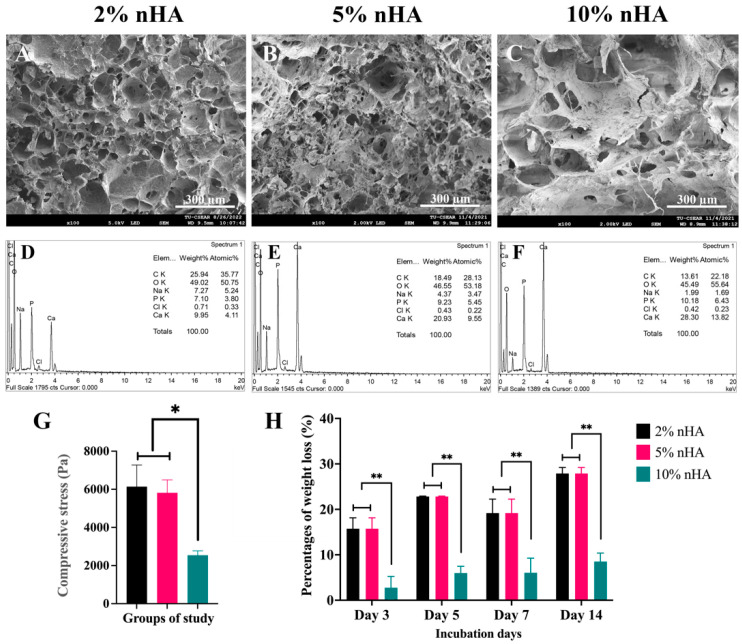
Effects of nanohydroxyapatite particles on microstructures, mineral compositions, mechanical strength, and physical stability of the thermosensitive nanohydroxyapatite–2% (*w*/*v*) calcium carbonate microcapsule–10% (*w*/*v*) β-glycerophosphate–4:1 (*w*/*w*) chitosan/collagen hydrogel (nHA–CaCO_3_–chitosan/collagen). (**A**–**C**) Scanning electron microscopy images and (**D**–**F**) energy dispersive X-ray spectroscopy (EDS) of the freeze-dried hydrogels: (**A**,**D**) 2% nHA, (**B**,**E**) 5% nHA, and (**C**,**F**) 10% nHA. (**G**) Compressive strength and (**H**) degradation rates of the hydrogels. The * and ** symbols represent significantly different values with *p* < 0.05 and *p* < 0.01, respectively (n = 5, mean ± SD).

**Figure 3 polymers-15-00416-f003:**
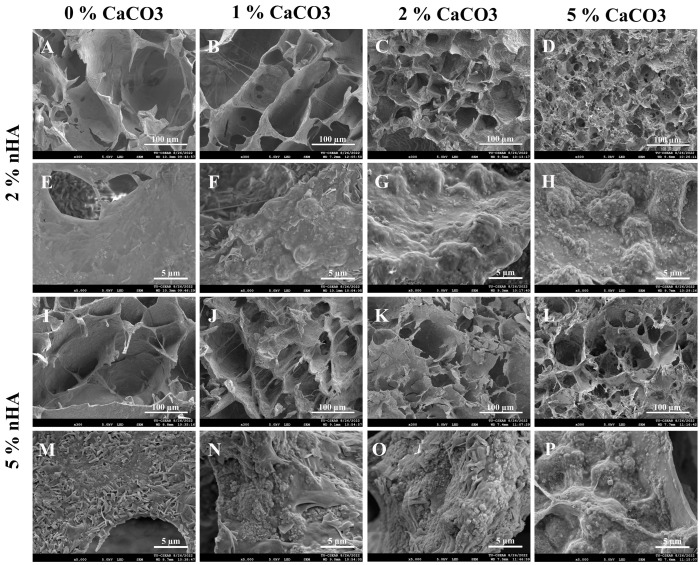
Scanning electron microscopy images exhibiting effects of calcium carbonate microcapsules (1%, 2%, and 5% (*w*/*v*)) on microstructures and pore walls of the thermosensitive calcium carbonate microcapsule–2% and 5% (*w*/*v*) nanohydroxyapatite–10% (*w*/*v*) β-glycerophosphate–4:1 (w/w) chitosan/collagen (nHA–CaCO_3_–chitosan/collagen) hydrogels. Hydrogels with (**A**,**E**,**I**,**N**) 0% CaCO_3_, (**B**,**F**,**J**,**N**) 1% CaCO_3_, (**C**,**G**,**K**,**O**) 2% CaCO_3_, and (**D**,**H**,**L**,**P**) 5% CaCO_3_. Hydrogels with (**A**–**H**) 2% nHA and (**I**–**P**) 5% nHA. The microstructures of the hydrogels are shown in (**A**–**D**,**I**–**L**) and surface features of pore walls (**E**–**H**,**M**–**P**). The small nHA particles are in nanometers, and the large spherical CaCO_3_ microcapsules are in micrometers. The lengths of the magnification bars are 100 and 5 µm.

**Figure 4 polymers-15-00416-f004:**
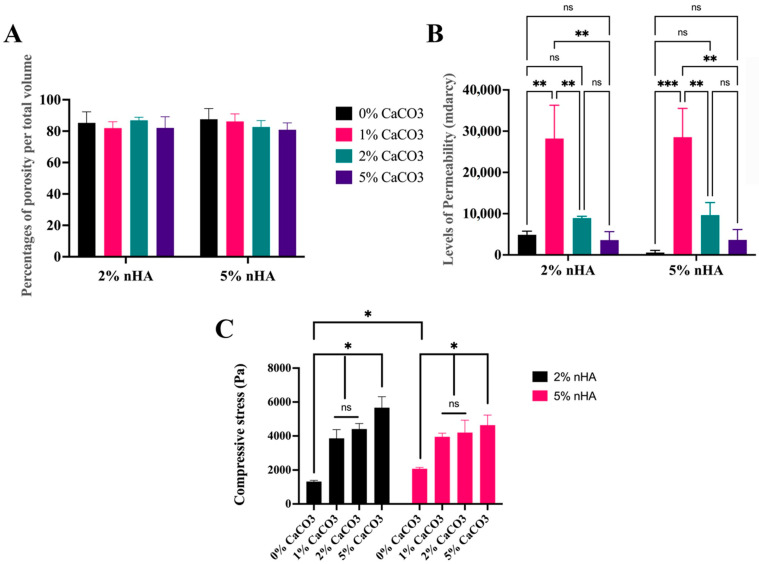
Mercury intrusion porosimetry (MIP) and compression tests demonstrating effects of calcium carbonate microcapsules on porous structures and mechanical strength of the thermosensitive 1%, 2%, and 5% calcium carbonate microcapsules–2% and 5% (*w*/*v*) nanohydroxyapatite–10% (*w*/*v*) β-glycerophosphate–4:1 (*w*/*w*) chitosan/collagen hydrogel (nHA–CaCO_3_–chitosan/collagen). MIP analysis on (**A**) percentages of total porosity and (**B**) permeability. (**C**) Compressive strength of the hydrogels. The abbreviation mdracy denotes millidarcy (1 Darcy is approximately 10^−12^ m^2^), and ns denotes “not significantly different at *p* > 0.05”. Symbols *, **, and *** represent significantly different values with *p* < 0.05, *p* < 0.01, and *p* < 0.001, respectively (n = 5, mean ± SD).

**Figure 5 polymers-15-00416-f005:**
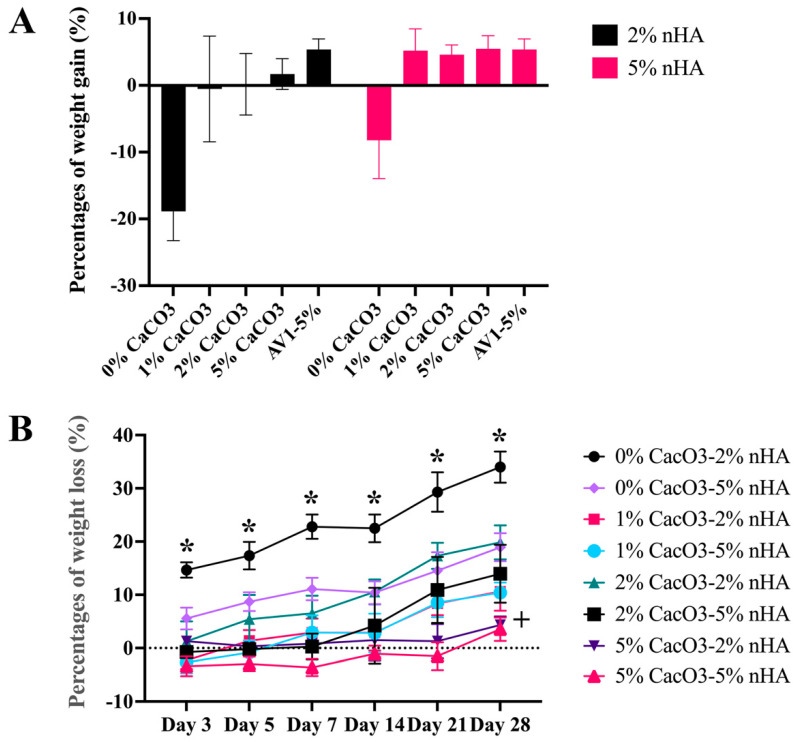
Effects of calcium carbonate microcapsules on physical properties of the 1%, 2%, and 5% calcium carbonate microcapsules-2% and 5% (*w*/*v*) nanohydroxyapatite-10% (*w*/*v*) β–glycerophosphate–4:1 (*w*/*w*) chitosan/collagen hydrogel (nHA–CaCO_3_–chitosan/collagen). The concentrations of CaCO_3_ were 1%, 2%, and 5%, and the concentrations of nHA were 2% and 5% (*w*/*v*), respectively. (**A**) Degrees of swelling of the hydrogels in phosphate buffer saline for 24 h and (**B**) degradation rates in a 1:1000 unit/mL lysozyme at 37 °C for 28 days. The symbol * represents significantly higher and lower values than other groups with *p* < 0.05 (n = 5, mean ± SD).

**Figure 6 polymers-15-00416-f006:**
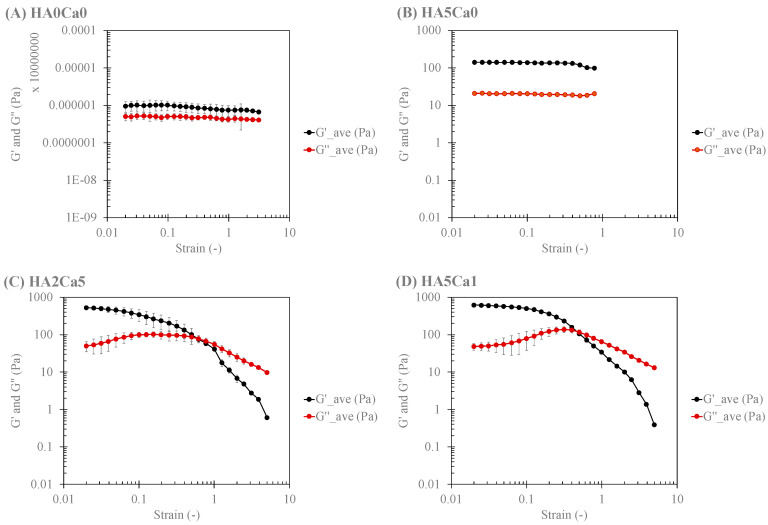
Amplitude (strain) dependence of four systems: System 1: (**A**) chitosan/collagen hydrogel alone; System 2: (**B**) hydrogel with 5% nHA; System 3: (**C**) hydrogel with 2% nHA and 5% CaCO_3_; and System 4: (**D**) hydrogel with 5% nHA and 1% CaCO_3_.

**Figure 7 polymers-15-00416-f007:**
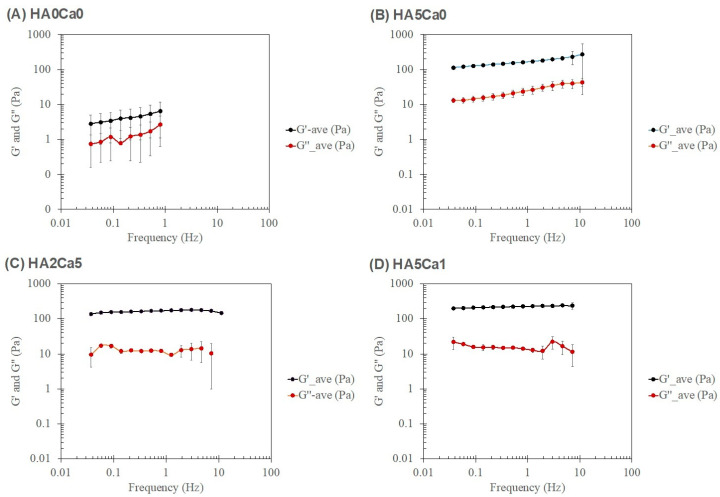
Frequency dependence of four systems: (**A**) System 1: chitosan/collagen hydrogel alone; (**B**) System 2: hydrogel with 5% nHA; (**C**) System 3: hydrogel with 2% nHA and 5% CaCO_3_; and (**D**) System 4: the hydrogel with 5% nHA and 1% CaCO_3_.

**Figure 8 polymers-15-00416-f008:**
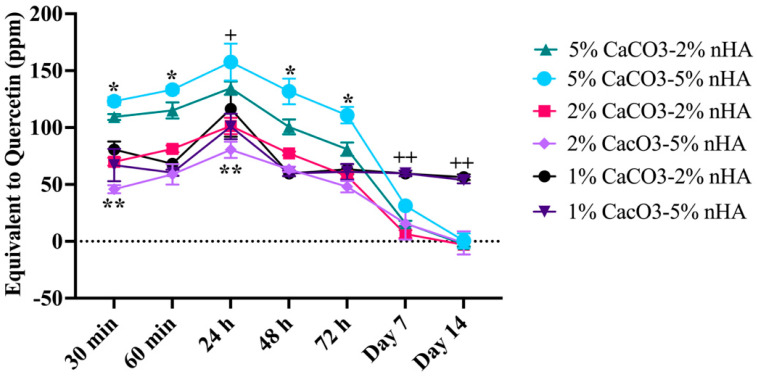
Total flavonoid content assay demonstrating effects of calcium carbonate microcapsules and nanohydroxyapatite particles on released profiles of QT from the thermosensitive calcium carbonate microcapsule–nanohydroxyapatite–10% (*w*/*v*) β–glycerophosphate–4:1 (*w*/*w*) chitosan/collagen hydrogels (nHA–CaCO_3_–chitosan/collagen). The concentrations of CaCO_3_ were 1%, 2%, and 5%, and nHA 2%, and 5% (*w*/*v*). The hydrogels were incubated in phosphate buffer saline at 37 °C for 14 days. Symbols * and ++ represent significantly higher and ** lower than other groups and + significantly higher than 2% CaCO_3_–5% nHA only at *p* < 0.05 (n = 5, mean ± SD).

**Figure 9 polymers-15-00416-f009:**
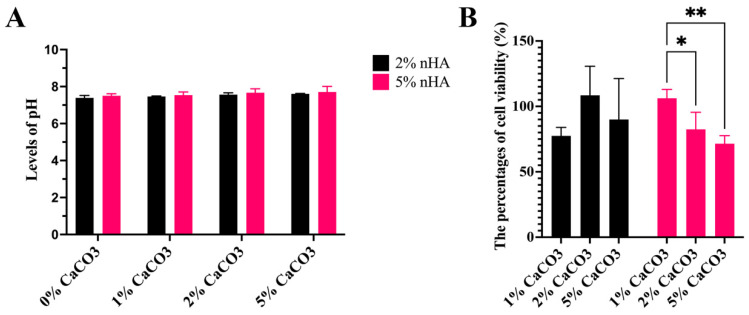
Effects of calcium carbonate microcapsules on acid–base balance (pH) and cell cytotoxicity of the thermosensitive calcium carbonate microcapsule–nanohydroxyapatite–10% (*w*/*v*) β-glycerophosphate–4:1 (*w*/*w*) chitosan/collagen (CaCO_3_–nHA–chitosan/collagen) hydrogels. The concentrations of CaCO_3_ were 1%, 2%, and 5%, and those of nHA were 2% and 5% (*w*/*v*). (**A**) Levels of pH and (**B**) percentages of cell viability derived from cell viability assay. The symbols * and ** represent significantly different values with *p* < 0.05 and *p* < 0.01 (n = 5, mean ± SD).

**Figure 10 polymers-15-00416-f010:**
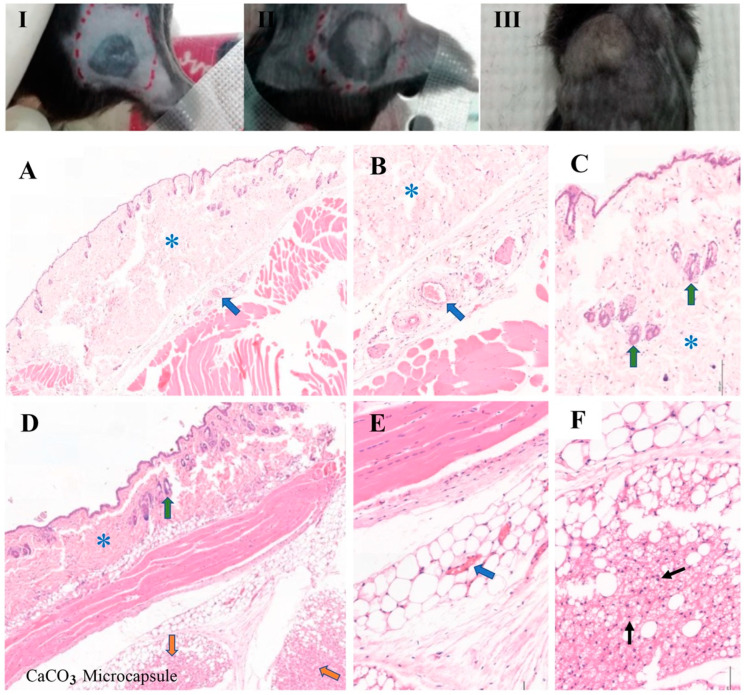
Subcutaneous implants of the thermosensitive calcium carbonate and 2% (*w*/*v*) calcium carbonate microcapsule–2% nanohydroxyapatite–10% β-glycerophosphate–4:1 (*w*/*w*) chitosan/collagen hydrogel in the C57BL/6NJcl mice for 4 weeks. (**I**–**III**) Before specimen harvesting, exhibiting normal skin and small nodules of the implant materials: (**I**) sham, (**II**) CaCO_3_, and (**III**) hydrogel groups. (**A**–**F**) Histological images of the (**A**–**C**) sham and (**D**–**E**) CaCO_3_ groups. (**A**,**B**,**E**) At the implant/tissue interface, demonstrating infiltration of small blood vessels in the thin fibrous connective tissue that lay between the skin and muscles layers (thick blue arrows) and (**F**) lymphocytes (thin black arrows) in the implantation site of the microcapsules group. Thick green arrows indicate hair follicles in the subcutaneous tissue of skin. Blue asterisks (*) demonstrated subcutaneous tissue of skin, and orange arrows indicate the implant site of the CaCO_3_ (the microcapsules cannot be seen). The thick–blue and thin black arrows suggested routine healing of the implantation sites in the sham and CaCO_3_ groups. (Hematoxylin and eosin, H&E stain), (**A**,**D**) at 10× magnification; (**B**,**C**,**E**,**F**) at 20× magnification).

**Figure 11 polymers-15-00416-f011:**
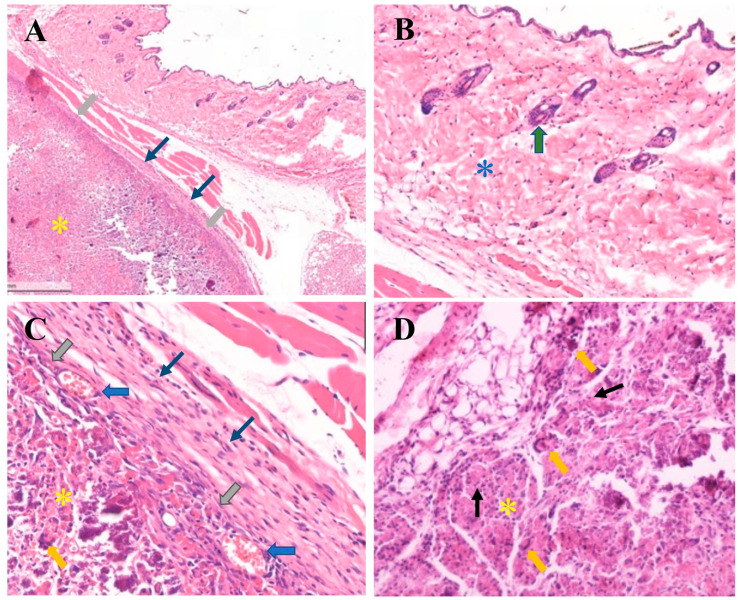
Histological images of the subcutaneous implants of a hydrogel group; the thermosensitive 2% (*w*/*v*) calcium carbonate microcapsule–2% nanohydroxyapatite–10%β-glycerophosphate–4:1 (*w*/*w*) chitosan/collagen hydrogels were subcutaneously implanted in C57BL/6NJcl mice for 4 weeks (**A**–**D**). (**A**) Spatial implant/tissue interface exhibiting that the sample is separated from the host tissue by thin layers of fibrous connective tissue (thin blue arrows), and thick grey arrows indicate inter-surface contact between the subcutaneous tissue and a residual implant or hydrogel (yellow asterisks). (**B**) A high magnification of mouse skin comprises dense fibrous connective tissue (blue asterisk), the hair follicle (thick green arrow), and keratinized epithelium. (**C**) A high magnification of a sample/tissue interface exhibiting multiple layers of connective tissue (thin blue arrows), direct contact of the sample, and the connective tissue layers (thick grey arrows) with infiltration of small blood vessels at the interface (thick blue arrows). (**D**) Exhibiting the residual implant or hydrogel (yellow asterisk), and low numbers of lymphocytes infiltration (chronic inflammatory cells) (thin black arrows) and multinucleated giant cells (thick yellow arrows) (hematoxylin and eosin, H&E stain), (**A**) magnification 10×; (**B**–**D**) 40×).

**Table 1 polymers-15-00416-t001:** Rheological parameters for the strain sweep test at the one hertz frequency of the hydrogels.

Sample Groups	G′@ LVR (Pa)	G″@ LVR (Pa)	Humpof G″	σ @′ = G″ (Pa)
System 1				
Chitosan/collagen hydrogel alone	9 ± 2	5 ± 1	None	None
System 2				
0%CaCO_3_-2%nHA	24 ± 1	8 ± 1	None	None
0%CaCO_3_-5%nHA	132 ± 7	20 ± 2	None	None
System 3				
1%CaCO_3_-2%nHA	418 ± 56	55 ± 6	Yes	26 ± 11
2%CaCO_3_-2%nHA	578 ± 181	65 ± 20	Yes	70 ± 48
5%CaCO_3_-2%nHA	495 ± 33	79 ± 11	Yes	48 ± 19
System 4				
1%CaCO_3_-5%nHA	600 ± 74	53 ± 8	Yes	62 ± 14
2%CaCO_3_-5%nHA	623 ± 95	89 ± 11	Yes	81 ± 8
5%CaCO_3_-5%nHA	568 ± 79	91 ± 9	Yes	76 ± 15

Note: G, storage modulus; G″, loss modulus; LVR, linear viscoelastic region; @, at; σ, strain; Pa, Pa.

**Table 2 polymers-15-00416-t002:** Rheological parameters of the power law model for the frequency sweep test at 1% strain and types of hydrogels.

Sample Groups	F (Power Law) (Hz)	Power Law Parameters				Characteristic Responses of the Hydrogels
		G′		G″		
		A′	n′	A″	n″	
System 1						
Chitosan/collagen hydrogel alone	0.2–1.0	3 ± 2	1.0 ± 0.2	0.8 ± 0.5	1.5 ± 0.1	Weak gel
System 2						
0%CaCO_3_-2%nHA	0.1–10.0	40 ± 12	0.07 ± 0.01	11 ± 3	0.09 ± 0.01	Weak–Strong gel
0%CaCO_3_-5%nHA	0.1–10.0	140 ± 20	0.06 ± 0.01	17 ± 3	0.09 ± 0.01	Strong gel
System 3						
1%CaCO_3_-2%nHA	0.4–1.2	137 ± 7	0.12 ± 0.02	11 ± 1	−0.1 ± 0.2	Strong particulate gel
2%CaCO_3_-2%nHA	0.4–3.0	151 ± 11	0.08 ± 0.01	11 ± 2	−0.1 ± 0.2	Strong particulate gel
5%CaCO_3_-2%nHA	0.4–4.6	330 ± 30	0.04 ± 0.01	26 ± 24	0.0 ± 0.1	Strong particulate gel
System 4						
1%CaCO_3_-5%nHA	0.2–1.9	202 ± 15	0.10 ± 0.02	19 ± 2	−0.3 ± 0.1	Strong particulate gel
2%CaCO_3_-5%nHA	0.2–7.2	193 ± 12	0.03 ± 0.02	17 ± 1	−0.1 ± 0.1	Strong particulate gel
5%CaCO_3_-5%nHA	0.6–7.2	226 ± 53	0.03 ± 0.01	17 ± 4	−0.1 ± 0.1	Strong particulate gel

Note: abbreviations: G′, storage modulus; G″, loss modulus; F, (power law); frequency range, power law; A′ and A″, power law constants of G′ and G″; n′ and n″, power law exponents of G′ and G″.

## Data Availability

The data that support the findings of this study are available from the corresponding author upon reasonable request.

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
