# Peer review of "Effects of Calcium Carbonate Microcapsules and Nanohydroxyapatite on Properties of Thermosensitive Chitosan/Collagen Hydrogels"

_polymers, 2023, doi:10.3390/polym15020416_

Round 1

Reviewer 1 Report

In this work, the authors prepared some complexes of CaCO3 and hydroxyapatite, tested their mechanical properties. The reviewer doesn't think the work is promising enough to be published in the high impact factor journal polymers, so the authors may submit it elsewhere. The following should also be considered before resubmission, in Figure 1b, the reason showing a broad peak in 24h should be given.

Author Response

Reviewer 1

Comments:

The manuscript should be revised. The following should also be considered before resubmission, in Figure 1b, the reason showing a broad peak in 24h should be given.

Responses:

  1. Introduction, Results and Conclusion are revised.
  2. Figure 1B shows burst release of quercetin from the calcium carbonate microcapsule after 1 hour incubation and a board peak in 24 h. This was because quercetin was physically absorbed on surface of the calcium carbonate particles and entrapped in the nano porous structure of the microcapsules and releasing rate was also related to concentrations of the substance[1,2]. Therefore, those non-covalent bonds of the molecules caused the burst release in the first hour. As the release process proceeded, the quercetin in nanoporous structure started to be released. Such simultaneous events caused a hump in release profile before the steady state was reach after 7 days. 
  3. Please see Section 4. Discussion, Page 22, Lines 622-629.

Reviewer 2 Report

Reviewer 1: I read the manuscript entitled as "Effects of calcium carbonate microcapsules and nanohydroxyapatite on properties of thermosensitive chitosan/collagen hydrogels". In my opinion the presented manuscript can be published in "Polymers" after minor revision. My comments are as follows:

1. Do not use abbreviation in Abstract.

2. Mention the novelty of work in the last paragraph of Introduction.

3. The authors need to proofread the manuscript due to grammatical errors

4. changes beta-glycerophosphate by β-glycerophosphate

5. Introduction need citations. Check through the complete manuscript and cite where it is needed.

6. I suggest to cite the following papers: DOI: 10.1016/j.matdes.2021.109865, DOI: 10.1016/j.cjac.2022.100092, DOI: 10.3389/fchem.2022.958420, DOI: 10.1155/2020/8747639, DOI: 10.1159/000485502, DOI: 10.1080/09205063.2022.2088528, DOI: 10.1201/9781003266518-11

7. Please improve the conclusion sections with major findings and suggestions for future research directions

Author Response

Reviewer 2

Comments:

  1. Do not use abbreviation in Abstract.

Responses:  Abbreviations are removed from the Abstract, on Page 1.

  1. Mention the novelty of work in the last paragraph of Introduction.

Responses:  The conceptual framework representing the novelty of the study is added in the last paragraph of Section 1. Introduction, and 5 new references were added in the introduction.

Please see Section 1. Introduction, Page 2, Lines 84 – 93 and Page 3, Lines 98 - 99.

List of five new references

  • Ma, P.; Wu, W.; Wei, Y.; Ren, L.; Lin, S.; Wu, J. Biomimetic gelatin/chitosan/polyvinyl alcohol/nano-hydroxyapatite scaffolds for bone tissue engineering. Materials & Design 2022, 207 109865, doi:https://doi.org/10.1016/j.matdes.2021.109865.
  • Sun, Q.; Yu, L.; Zhang, Z.; Qian, C.; Fang, H.; Wang, J.; Wu, P.; Zhu, X.; Zhang, J.; Zhong, L.; et al. A novel gelatin/carboxymethyl chitosan/nano-hydroxyapatite/beta-tricalcium phosphate biomimetic nanocomposite scaffold for bone tissue engineering applications. Front Chem 2022, 10, 958420, doi:10.3389/fchem.2022.958420.
  • Balestrin, L.A.; Back, P.I.; Marques, M.D.S.; Araujo, G.M.S.; Carrasco, M.C.F.; Batista, M.M.; Silveira, T.; Rodrigues, J.L.; Fachel, F.N.S.; Koester, L.S.; et al. Effect of Hydrogel Containing Achyrocline satureioides (Asteraceae) Extract-Loaded Nanoemulsions on Wound Healing Activity. Pharmaceutics 2022, 14, doi:10.3390/pharmaceutics14122726.
  • Chen, M.; Li, M.; Wei, Y.; Xue, C.; Chen, M.; Fei, Y.; Tan, L.; Luo, Z.; Cai, K.; Hu, Y. ROS-activatable biomimetic interface mediates in-situ bioenergetic remodeling of osteogenic cells for osteoporotic bone repair. Biomaterials 2022, 291, 121878, doi:10.1016/j.biomaterials.2022.121878.
  • Yang, Y.; Chen, D.; Li, Y.; Zou, J.; Han, R.; Li, H.; Zhang, J. Effect of Puerarin on Osteogenic Differentiation in vitro and on New Bone Formation in vivo. Drug Des Devel Ther 2022, 16, 2885-2900, doi:10.2147/DDDT.S379794.
  1. The authors need to proofread the manuscript due to grammatical errors

Responses:  The manuscript was re-proofread by a professional language service.

  1. Changes beta-glycerophosphate by β-glycerophosphate

Responses:  beta-glycerophosphate was changed to β-glycerophosphate.

  1. Introduction need citations. Check through the complete manuscript and cite where it is needed.

Responses: A total of 10 new references including the recommended articles are added in Section 1. Introduction, Reference numbers 7, 8, 28, 30 and 31 on Page 2,  and Section 4. Discussion, Reference numbers 28 and 46-49, on Pages 22-24.

  1. Please improve the conclusion sections with major findings and suggestions for future research directions

Responses:  Section 5. Conclusion is re-written as suggested

Please see Section 5 Conclusion, on Page 24, Lines 711 -729.

  1. List of a total 10 new references (Cited reference numbers Reference numbers 7, 8, 26, 28, 30, 31, 46-49)
  • Ma, P.; Wu, W.; Wei, Y.; Ren, L.; Lin, S.; Wu, J. Biomimetic gelatin/chitosan/polyvinyl alcohol/nano-hydroxyapatite scaffolds for bone tissue engineering. Materials & Design 2022, 207 109865, doi:https://doi.org/10.1016/j.matdes.2021.109865.
  • Sun, Q.; Yu, L.; Zhang, Z.; Qian, C.; Fang, H.; Wang, J.; Wu, P.; Zhu, X.; Zhang, J.; Zhong, L.; et al. A novel gelatin/carboxymethyl chitosan/nano-hydroxyapatite/beta-tricalcium phosphate biomimetic nanocomposite scaffold for bone tissue engineering applications. Front Chem 2022, 10, 958420, doi:10.3389/fchem.2022.958420.
  • Balestrin, L.A.; Back, P.I.; Marques, M.D.S.; Araujo, G.M.S.; Carrasco, M.C.F.; Batista, M.M.; Silveira, T.; Rodrigues, J.L.; Fachel, F.N.S.; Koester, L.S.; et al. Effect of Hydrogel Containing Achyrocline satureioides (Asteraceae) Extract-Loaded Nanoemulsions on Wound Healing Activity. Pharmaceutics 2022, 14, doi:10.3390/pharmaceutics14122726.
  • Chen, M.; Li, M.; Wei, Y.; Xue, C.; Chen, M.; Fei, Y.; Tan, L.; Luo, Z.; Cai, K.; Hu, Y. ROS-activatable biomimetic interface mediates in-situ bioenergetic remodeling of osteogenic cells for osteoporotic bone repair. Biomaterials 2022, 291, 121878, doi:10.1016/j.biomaterials.2022.121878.
  • Yang, Y.; Chen, D.; Li, Y.; Zou, J.; Han, R.; Li, H.; Zhang, J. Effect of Puerarin on Osteogenic Differentiation in vitro and on New Bone Formation in vivo. Drug Des Devel Ther 2022, 16, 2885-2900, doi:10.2147/DDDT.S379794.
  • Kalmykova, T.P.; Kostina, Y.V.; Ilyin, S.O.; Bogdanova, Y.G.; Severin, A.V.; Ivanov, P.L.; Antonov, S.V. Effect of Synthesis Medium on the Structure and Physicochemical Properties of Biomineral Composites Based on Hydroxyapatite and Hyaluronic Acid. Polymer Science, Series B 2020, 62, 61-71, doi:10.1134/S1560090420010042.
  • Hassani, L.N.; Hindre, F.; Beuvier, T.; Calvignac, B.; Lautram, N.; Gibaud, A.; Boury, F. Lysozyme encapsulation into nanostructured CaCO(3) microparticles using a supercritical CO(2) process and comparison with the normal route. J Mater Chem B 2013, 1, 4011-4019, doi:10.1039/c3tb20467g.
  • Schliephake, H. Application of bone growth factors--the potential of different carrier systems. Oral Maxillofac Surg 2010, 14, 17-22, doi:10.1007/s10006-009-0185-1.
  • Tan, C.; Dima, C.; Huang, M.; Assadpour, E.; Wang, J.; Sun, B.; Kharazmi, M.S.; Jafari, S.M. Advanced CaCO(3)-derived delivery systems for bioactive compounds. Adv Colloid Interface Sci 2022, 309, 102791, doi:10.1016/j.cis.2022.102791.
  • Niu, X.; Feng, Q.; Wang, M.; Guo, X.; Zheng, Q. Porous nano-HA/collagen/PLLA scaffold containing chitosan microspheres for controlled delivery of synthetic peptide derived from BMP-2. J Control Release 2009, 134, 111-117, doi:10.1016/j.jconrel.2008.11.020.

Reviewer 3 Report

The article by Arpornmaeklong P. et al. investigates gels based on a cross-linked mixed aqueous solution of chitosan and collagen with the addition of hydroxyapatite and quercetin-containing calcium carbonate particles. The authors position their samples for regenerative medicine and investigate their properties in the complex: from morphology and rheology to cytotoxicity and histological evaluation of the biocompatibility. The article is interesting and multifaceted but requires an editorial revision and correction of some places listed below. In addition, the authors use two words that without proof look speculative. First, they write that they use "nanohydroxyapatite," but provide no evidence that it is actually nanoscale particles. Second, they write about the "thermosensitive" gel, but provide no proof as well. Thermosensitivity of a gel means that reversible transitions from sol to gel and, vice versa, from gel to sol are possible. Based on the description of the article, the authors carry out the transition of sol to gel by chemical crosslinking of dissolved macromolecules, which means that the reverse transition of gel to sol on cooling is not possible and the gel is not thermosensitive. In that case, the word "thermosensitive" should be removed. These points should be taken into account when preparing a revised version of the manuscript.

Other specific comments are as follows.

Page 2: “1 mg/mL carboxymethylcellulose under stirring”. The molecular weight of carboxymethylcellulose (or its intrinsic viscosity) and the degree of substitution must be specified.

Page 3: “of 2% chitosan in 0.1 M”. The molecular weight of chitosan (or its intrinsic viscosity) and the degree of substitution must be specified.

Page 3, Section 2.2: The section should provide information on the synthesis of nanohydroxyapatite, as well as its particle size and the method of its determination. It would also be useful to show a SEM or TEM image.

Page 8, Figure 2A-C: A clearer dimensional scale with its size indication should be given on the SEM images.

Fig. 3. There is no point in specifying a magnification for SEM images, because the size of the image changes when it is scaled in the journal. The authors must provide clear dimensional scales on the figures with indication of their size.

Page 9: “28,212.69 ± 8074.03 mdacry” and other values, Table 1, Table 2, and everywhere else: Values should be rounded according to their standard deviation, i.e. “28,212.69 ± 8074.03 mdacry” -> “28,000 ± 8000 mdacry”, “3619.86 ± 28.57” -> “3620 ± 30”; “4034.59 ± 242.58” -> “4030 ± 240”, and so on.

Page 10: “1 Darcy is approximately 10–12 m” -> “1 Darcy is approximately 10–12 m2”.

Page 11: “3.8. Rheological property”. There are numerous rheological properties. The word "property" should be used in the plural, or, alternatively, the word "characteristics" should be used.

Page 12. The authors should provide figures with amplitude dependences of the storage and loss moduli from which the data in Table 1 were extracted. For example, I do not understand why there is no yield point for gels without calcium carbonate in Table 1. In addition, it is not clear what "F (power law) (Hz)" is and why it has such strange values in Table 2.

Tables 1, 2: “CaCO” -> “CaCO3”.

Page 12: "System 1 exhibited the property of a weak gel, in which the values of the power law exponent (n’ and n’’) were greater than 0." Judging by the values, it is not a "weak gel" but a regular polymer solution. The authors should provide figures with frequency dependences of storage and loss moduli for all systems. In addition, it is known that the addition of Ca-based microparticles to biopolymer solutions leads to the formation of gels (see, e.g., 10.1134/S1560090420010042), and this should be mentioned. In other words, the gel can form not only due to chemical cross-linking of macromolecules but also due to their interaction with dispersed particles and formation of physical cross-links.

Page 14: “106.26 ± 6.35%” I do not understand how percentages of cell viability can be greater than 100%. The authors should explain this.

Author Response

Reviewer 3

Comments:

General comments

The article by Arpornmaeklong P. et al. investigates gels based on a cross-linked mixed aqueous solution of chitosan and collagen with the addition of hydroxyapatite and quercetin-containing calcium carbonate particles. The authors position their samples for regenerative medicine and investigate their properties in the complex: from morphology and rheology to cytotoxicity and histological evaluation of the biocompatibility. The article is interesting and multifaceted but requires an editorial revision and correction of some places listed below. In addition, the authors use two words that without proof look speculative.

First, they write that they use "nanohydroxyapatite," but provide no evidence that it is actually nanoscale particle.

Second, they write about the "thermosensitive" gel, but provide no proof as well. Thermosensitivity of a gel means that reversible transitions from sol to gel and, vice versa, from gel to sol are possible. Based on the description of the article, the authors carry out the transition of sol to gel by chemical crosslinking of dissolved macromolecules, which means that the reverse transition of gel to sol on cooling is not possible and the gel is not thermosensitive. In that case, the word "thermosensitive" should be removed. These points should be taken into account when preparing a revised version of the manuscript.

Responses

First:       nanohydroxyapatite," but provide no evidence that it is actually nanoscale particle.

Responses: Nanohydroxyapatite used in this study was a commercial nano powder size 100 – 200 µm (BET) obtained from Sigma-Aldrich, described in Section 2. Materials and Methods, 2.1 Materials, Page 3, Lines 108-109.

Second:   "thermosensitive" gel, but provide no proof as well and the word should be removed.

Responses:

  1. A method to determine sol-gel transition time and results demonstrated phase changes from sol to gel were revised and elaborated, in Section 2. Materials and methods, Sub-section 2.3. Preparation of thermosensitive hydrogel, on Page 4, Lines 153-156, and Section 3. Results, Subsection 3.4. Effects of calcium carbonate microcapsules on sol-gel transition time and microstructure of the hydrogels, on Page 9, Lines 359, 362 – 366, “The study groups were 1%, 2%, and 5% CaCO3 in the 2% and 5% nHA-hydrogels (6 groups). The hydrogels were in liquid stage at 4°C and became solid and opaque white gel after incubation at 37°C for 7 – 15 min depending on the concentrations of the CaCO3 and nHA in the hydrogels. In 2% nHA hydrogels with 1%, 2% and 5% CaCO3, the setting times were in a range of 15 – 10 min, while the 5% nHA were 10 – 7 min, respectively.”  
  2. Word “Thermosensitive” is used in previous reports to describe the hydrogel that response to the temperature change from sol to get at 37ºC or body temperature and the physical bonds of those hydrogels are stimulated by beta-glycerophosphate and temperature change. They are thermosensitive, and the sol-gel transition cannot be reversed to solution stage, thus they are not a thermoreversible hydrogel.[1-4] Therefore, the word ‘Thermosensitive” is used in the current study. 
  3. Additionally, the method used to examine gel mobility is emphasized in Section 2. Materials and methods, Subsection 2.3 Preparation of the hydrogel, on Page 4, Lines 153 - 156, and the results of the setting time at 37°C was added in Section 3. Results, Subsection 3.4 Effects of calcium carbonate microcapsule, on Page 9, Lines 359, 362 -366.

  1. List of references for thermosensitive hydrogels
  • Cheng, Y.H.; Ko, Y.C.; Chang, Y.F.; Huang, S.H.; Liu, C.J. Thermosensitive chitosan-gelatin-based hydrogel containing curcumin-loaded nanoparticles and latanoprost as a dual-drug delivery system for glaucoma treatment. Exp Eye Res 2019, 179, 179-187, doi:10.1016/j.exer.2018.11.017.
  • 2 Dhivya, S.; Saravanan, S.; Sastry, T.P.; Selvamurugan, N. Nanohydroxyapatite-reinforced chitosan composite hydrogel for bone tissue repair in vitro and in vivo. J Nanobiotechnology 2015, 13, 40, doi:10.1186/s12951-015-0099-z.
  • Moreira, C.D.; Carvalho, S.M.; Mansur, H.S.; Pereira, M.M. Thermogelling chitosan-collagen-bioactive glass nanoparticle hybrids as potential injectable systems for tissue engineering. Mater Sci Eng C Mater Biol Appl 2016, 58, 1207-1216, doi:10.1016/j.msec.2015.09.075.
  • Wang, L.; Stegemann, J.P. Thermogelling chitosan and collagen composite hydrogels initiated with beta-glycerophosphate for bone tissue engineering. Biomaterials 2010, 31, 3976-3985, doi:10.1016/j.biomaterials.2010.01.131.

Other specific comments are as follows.

  1. Page 2: “1 mg/mL carboxymethylcellulose under stirring”. The molecular weight of carboxymethylcellulose (or its intrinsic viscosity) and the degree of substitution must be specified.
  2. Page 3: “of 2% chitosan in 0.1 M”. The molecular weight of chitosan (or its intrinsic viscosity) and the degree of substitution must be specified.

Responses:  Detail properties of the chemicals are added in a new section, Subsection 3.1 Material, on Page 3, Lines 103 – 105 as follows, “…, carboxymethylcellulose (average molecular weight (MW) ~90,000 with 0.7 carboxymethyl groups per anhydroglucose unit), chitosan (medium molecular weight, 75-85% deacetylated),…”.

  1. Page 3, Section 2.2: The section should provide information on the synthesis of nanohydroxyapatite, as well as its particle size and the method of its determination. It would also be useful to show a SEM or TEM image.

Responses:  The nanohydroxyapatite particles are a commercial nano powder obtained from Sigma-Aldrich and details are explained in Subsection 3.1 Material, on Page 3, Lines 107 – 109 as follows, “, hydroxyapatite nanopowder (< 200 nm particle size (BET), ≥ 97% synthetic),”

  1. Page 8, Figure 2A-C: A clearer dimensional scale with its size indication should be given on the SEM images, and
  2. 3. There is no point in specifying a magnification for SEM images, because the size of the image changes when it is scaled in the journal. The authors must provide clear dimensional scales on the figures with indication of their size.

Responses: Size of dimension scales of Figure 2A-C and Figure 3A-Q are increased, in proportion to the original width of the scale on the original SEM images, as shown in Section 3. Results, on Pages 9 and 10, and “(magnification x 300) and (magnification x 5000) on Figure 3, Page 10, Line 381 were removed.

  1. Page 9: “28,212.69 ± 8074.03 mDacry” and other values, Table 1, Table 2, and everywhere else: Values should be rounded according to their standard deviation, i.e. “28,212.69 ± 8074.03 mDacry” -> “28,000 ± 8000 mDacry”, “3619.86 ± 28.57” -> “3620 ± 30”; “4034.59 ± 242.58” -> “4030 ± 240”, and so on.

Responses:  All numbers in the manuscript are rounded according to their standard deviation and highlighted.

  1. Page 10: “1 Darcy is approximately 10–12 m” -> “1 Darcy is approximately 10–12 m2”.

Responses: “… 10-12 m” is changed to “… 10-12 m2” in the figure legend of Figure 4 on Page 11, Line 409.

  1. Page 11: “8. Rheological property”. There are numerous rheological properties. The word "property" should be used in the plural, or, alternatively, the word "characteristics" should be used.

Responses: Title of section “3.8 Rheological property…” was renamed to “3.8 Rheological Characteristics…” according to the reviewer’s suggestion, Subsection 3.8 Rheological characteristics, on Page 13, Line 442.

  1. Page 12. The authors should provide figures with amplitude dependences of the storage and loss moduli from which the data in Table 1 were extracted. For example, I do not understand why there is no yield point for gels without calcium carbonate in Table 1.

Responses: Figures of amplitude (strain) dependence of storage and loss moduli of representatives of all the four systems were added (Figure 6, Page 14) and the relevant details were included accordingly (Line 447 – 451). The Figure 6 (A) and (B) show that G¢ and G² of the systems without CaCO3 microcapsules, i.e. chitosan/collagen hydrogel and the hydrogel with 5% nHA, were amplitude independent within the torque limit. Such behavior may imply that those gels are “strong” of which yield point exceeding 1 strain unit (Kavanagh & Ross-Murphy (1998)) or may undergo strain stiffening at large strains (Hyun et al. (2011) and Sim et al. (2003)).  

References:

  • Kavanagh, G. M., & Ross-Murphy, S. B. (1998). Rheological characterisation of polymer gels. Progress in Polymer Science23(3), 533-562.
  • Hyun, K., Wilhelm, M., Klein, C. O., Cho, K. S., Nam, J. G., Ahn, K. H., ... & McKinley, G. H. (2011). A review of nonlinear oscillatory shear tests: Analysis and application of large amplitude oscillatory shear (LAOS). Progress in Polymer Science36(12), 1697-1753.
  • Sim, H. G., Ahn, K. H., & Lee, S. J. (2003). Large amplitude oscillatory shear behavior of complex fluids investigated by a network model: A guideline for classification. Journal of Non-Newtonian Fluid Mechanics112(2-3), 237-250.
  1. Tables 1, 2: “CaCO” -> “CaCO3

Responses: All the words of “CaCO” were changed to “CaCO3(Table 1, Page 15 and Table 2, Page 17)

  1. Page 12: "it is not clear what "F (power law) (Hz)" is and why it has such strange values in Table 2. System 1 exhibited the property of a weak gel, in which the values of the power law exponent (n’ and n’’) were greater than 0." Judging by the values, it is not a "weak gel" but a regular polymer solution. The authors should provide figures with frequency dependences of storage and loss moduli for all systems. In addition, it is known that the addition of Ca-based microparticles to biopolymer solutions leads to the formation of gels (see, e.g., 10.1134/S1560090420010042), and this should be mentioned. In other words, the gel can form not only due to chemical cross-linking of macromolecules but also due to their interaction with dispersed particles and formation of physical cross-links.

Responses: The studied systems are rather complex thus the curves of G¢ and G² may not fall into a single power law. Thus, “F(power law)” was included in the Table 2 to indicate the frequency range of the data that can be fit by the power law relationship. The description of F(power law) was stated in the Note at the end of the table.  Figures of frequency dependence of storage and loss moduli of representatives of all the four systems, Figure 7 and Figure legend were added (Page 16) and the details was included accordingly (Page 16, Line 494-496). In Figure 7 (A) G¢ and G² of the chitosan/collagen hydrogel (System 1) shows power-law dependence on frequency throughout the studied frequency range. This rheological behavior is a characteristic of “weak gel” (Ikeda & Nishinari (2001) and Song et al. (2006)). Moreover, terminal zone of polymer solutions where G² ~ w > G¢ ~ w2 at low frequencies was not observed in our case.

References  

  • Ikeda, S., & Nishinari, K. (2001). “Weak gel”-type rheological properties of aqueous dispersions of nonaggregated κ-carrageenan helices. Journal of Agricultural and Food Chemistry49(9), 4436-4441.
  • Song, K. W., Kim, Y. S., & Chang, G. S. (2006). Rheology of concentrated xanthan gum solutions: Steady shear flow behavior. Fibers and Polymers7(2), 129-138.

  1. Page 14: “106.26 ± 6.35%” I do not understand how percentages of cell viability can be greater than 100%. The authors should explain this.

Responses:  The percentages of cell viability were calculated relatively to optical density (OD) or levels of metabolic activity of cells in a regular growth medium of a control group.  The percentages of cell viability that were higher than 100% indicated that growth of cells in the experimental groups were relatively higher than that of the control group.  A calculation method and formula are added in Section 2. Materials and methods, Subsection 2.11. Cytotoxicity test of the hydrogel, on Page 6, Lines 258 – 261.

Reviewer 4 Report

Journal: MDPI - Polymers

Manuscript

Title:  "Effects of calcium carbonate microcapsules and nanohydroxyapatite on properties of thermosensitive chitosan/collagen hydrogels"

Author(s): Premjit Arpornmaeklong, Natthaporn Jaiman, Komsan Apinyauppatham, Asira Fuongfuchat and Supakorn Boonyuen

Reviewer Comments to Author(s)

Recommendation: Minor Revisions

This manuscript presents a comprehensive study of chitosan/collagen hydrogels incorporating calcium carbonate microcapsules and nanohydroxyapatite. The manuscript corresponds to a well-structured research work presented. The author(s) could think of the following simple correctiors.

1.      Materials and Methods:

- The author(s) can add a part of Materials before paragraph 2.1 to present all the reagents, companies, purity used.

- the hydrogels in paragraph 2.2 were categorized in two systems with 2% and 5% nHA. Is there a 3rd system with 10% nHA as mentioned in line 110?

-line 174 “The hydrogel disks, 20 5 mm in size,…” probable the author(s) mean 20x5 mm?

-in paragraph 2.11 line 228 the author(s) state that eight male mice were purchased, but at lines 239-240they describe five samples per group A and B and three samples per group C, that is 13 samples (male mice). Please specify the correct number of mice or rephrase to provide the correct meaning.

instead of 10.

2.     Results:

-in paragraph 3.5, line 335 the content 1% CaCO3 has been mentioned twice.

-in paragraph 3.6, line 344 probable the author(s) wanted to mention 5% CaCO3

- in paragraph 3.7, lines 367-368 the concentration 0% CaCO3–2% nHA is referred twice (0% CaCO3–2% nHA > 0% CaCO3–2% nHA), is there a reason for this comparison?

- in paragraph 3.8, line 415 please correct the CaCO3 with the number as subscript, and at Table 2 last column please correct the characteristic response of 0%CaCO-2%nHA

- Figure 9, line 503, please correct “think”

Author Response

Reviewer 4:

Comments

  1. Materials and Methods: The author(s) can add a part of Materials before paragraph 2.1 to present all the reagents, companies, purity used.

Responses: Subsection 2.1 Materials is added on Page 3, Lines 102 -119

  1. The hydrogels in paragraph 2.2 were categorized in two systems with 2% and 5% nHA. Is there a 3rd system with 10% nHA as mentioned in line 110?

Responses: Following the results in Subsection 3.2. (Page 8, Lines 328-340), Effects of the nanohydroxyapatite on microstructure and porosity of the thermosensitive calcium carbonate microcapsules-nanohydroxyapatite-chitosan/collagen hydrogel and Subsection 3.3. Mechanical strength and degree of degradation of the thermosensitive calcium carbonate microcapsules-nanohydroxyapatite-chitosan/collagen hydrogel(Page 8, Line 341-346) that compared properties of 2%, 5% and 10% nHA-2% CaCO3 – hydrogels, the 2% and 5% nHA were selected for further study.  Therefore, for the analysis of the effects of the 1%, 2% and 5% CaCO3 microcapsules on property of the hydrogels, there were only two systems (Systems 1 and 2), based on 2%nHA and 5% nHA systems, on Page 8, Lines 345-346.

  1. Line 174 “The hydrogel disks, 20 5 mm in size,…” probable the author(s) mean 20x5 mm?

Responses: size of the hydrogel disks was changed from “20 5 mm in size” to “10 x 10 mm in size”, Subsection 2.8 Swelling test, on Page 5, Line 214.

  1. In paragraph 2.11 line 228 the author(s) state that eight male mice were purchased, but at lines 239-240they describe five samples per group A and B and three samples per group C, that is 13 samples (male mice). Please specify the correct number of mice or rephrase to provide the correct meaning. instead of 10.

Responses:

On Subsection 2.12 Subcutaneous implantation in mice, on Page 6, Line 261

  • Numbers of mice are changed to 7 mice (Line 268). A total number of mice that were ordered was 8 mice.  One extra mouse was ordered to covered for an unexpected loss.
  • The descriptions of the section are revised, that there are two pockets for two groups of samples on 6 mice and one mouse, there was only one pocket. Therefore, a total number of mice that were operated was 7 mice, 6 mice had 2 pockets and one mouse and one pocket (Line 279 – 282).
  1. Section 3. Results: in paragraph 3.5, line 335 the content 1% CaCO3 has been mentioned twice.

Responses: a repeated 1% CaCO3 is removed, Subsection 3.5 Mercury intrusion porosimetry, Subsection 3.5 Mercury intrusion porosimetry, Page 10, Line 389-391.

  1. in paragraph 3.6, line 344 probable the author(s) wanted to mention 5% CaCO3

Responses: “10% CaCO3” was changed to “5% CaCO3”, Subsection 3.6 . Mechanical strength, Page 11, Line 398

  1. in paragraph 3.7, lines 367-368 the concentration 0% CaCO3–2% nHA is referred twice (0% CaCO3–2% nHA > 0% CaCO3–2% nHA), is there a reason for this comparison?

Responses: a repeated “0% CaCO3–2% nHA” is removed, Subsection 3.7 Physical property, Page 12, on Line 424-425.

  1. in paragraph 3.8, line 415 please correct the CaCO3 with the number as subscript, and at Table 2 last column please correct the characteristic response of 0%CaCO-2%nHA

Responses: 

  • “CaCO3” is changed to “CaCO3” in Subsection 3.8 Rheological characteristics, on Page 16, Line 489 and
  • The “0%CaCO-2%nHA” is changed to “0%CaCO3-2%nHA” in Table2, last column, “Weak-Strong gel” on Page 17.
  1. Figure 9, line 503, please correct “think”

Responses:  Because two new figures (Figures 6 and 7) are added, Figure 9 is changed to Figure 11, on Page 21.  Therefore, “think” in the Figure legend of Figure 11 is changed to “thin”, on Page 21, Line 585.

Round 2

Reviewer 1 Report

na

Reviewer 3 Report

The authors have made great efforts to improve their manuscript and respond to the reviewer comments. Although I still do not like that the authors call their gels thermosensitive, I see no reason why this article cannot be published.